# The Empirical Impact of Data Sanitization on Language Models

Anwesan Pal[*], Radhika Bhargava[*], Kyle Hinsz, Jacques Esterhuizen, and Sudipta Bhattacharya

Amazon Web Services
{anwesanp, radhikb, khinsz, jesterhu, sudibhat}@amazon.com

## Abstract

Data sanitization in the context of language modeling involves identifying sensitive content, such as personally identifiable information (PII), and redacting them from a dataset corpus. It is a common practice used in natural language processing (NLP) to maintain privacy. Nevertheless, the impact of data sanitization on the language understanding capability of a language model remains less studied. This paper empirically analyzes the effects of data sanitization across several benchmark language-modeling tasks including comprehension question answering (Q&A), entailment, sentiment analysis, and text classification. Our experiments cover a wide spectrum comprising finetuning small-scale language models, to prompting large language models (LLMs), on both original and sanitized datasets, and comparing their performance across the tasks. Interestingly, our results suggest that for some tasks such as sentiment analysis or entailment, the impact of redaction is quite low, typically around 1-5%, while for tasks such as comprehension Q&A there is a big drop of >25% in performance observed in redacted queries as compared to the original. For tasks that have a higher impact, we perform a deeper dive to inspect the presence of task-critical entities. Finally, we investigate correlation between performance and number of redacted entities, and also suggest a strategy to repair an already redacted dataset by means of content-based subsampling. Additional details are available at https://sites.google.com/view/datasan.

## 1 Introduction

Data privacy is a critical concern in the development and use of language models (LMs) specially due to the sensitive nature of personally identifiable information (PII) that can be present in the text. PII commonly includes sensitive information such as person names, addresses, emails, or social security numbers. Data privacy concerns are in part motivated by security issues that arise from LMs memorizing portions of the training data, which can then be extracted via adversarial attacks [4, 26]. PII data breaches are a serious concern for large corporations, as they can lead to severe damage to the reputation and finances of an organization. Furthermore, corporate data governance policies are driven by applicable privacy laws which place strict legal limitations on the use of PII.

Some popular techniques to anonymize data in the Natural Language Processing (NLP) domain include differential privacy and data sanitization. Differential privacy [10] involves development of a mathematical framework that adheres to a rigorous definition of privacy by injecting noise into the data. While this guarantees that a trained model will not reveal any user identifiable information, adding noise comes with the price of great loss in data fidelity, which is not ideal for studying the impact of anonymization on model performance [23]. On the other hand, data sanitization

---

[*]Equal contribution

Safe Generative AI workshop at 38th Conference on Neural Information Processing Systems (NeurIPS 2024).

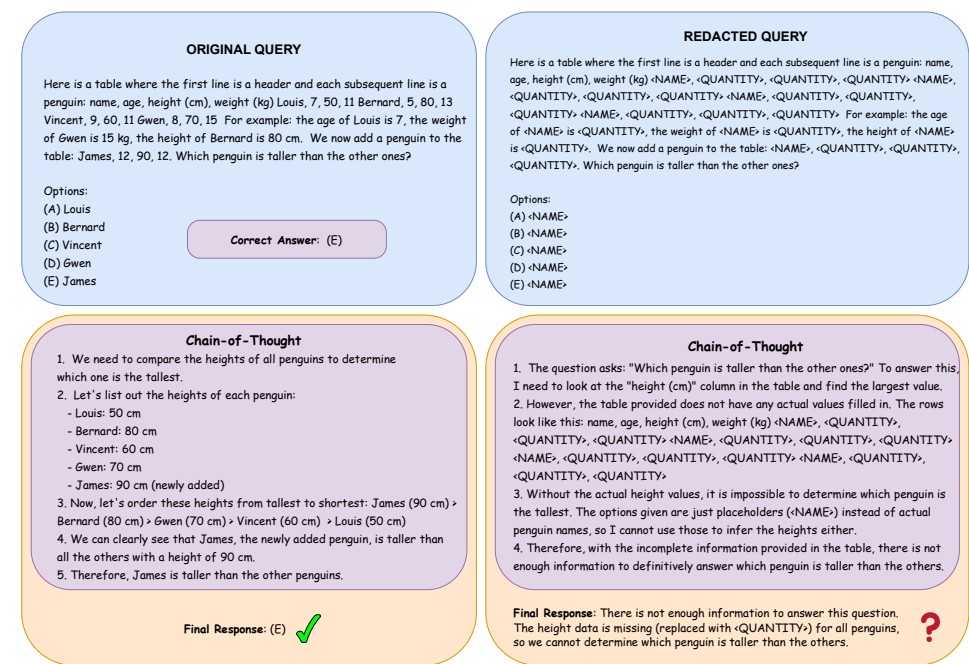

Figure 1: LLM chain-of-thought for a query in the original vs redacted forms. In the redacted case, the reasoning module correctly highlights missing information in the query, and is therefore unable to provide the answer to an otherwise straight-forward question.

involves complete and irreversible removal of personally identifiable information from data without the introduction of additional noise. Such a masking approach ensures that information pertaining to an individual cannot be recovered, either directly or in collaboration with a third party. In recent years, a number of organizations like Microsoft, Paypal, and Mastercard [2, 13, 38] have employed sanitized data for training LMs to leverage information present in their free text corpora while minimizing data leakage and privacy violations. Therefore, in this work, we adopt the data sanitization approach for analyzing impact on performance of language models via redaction.

Despite the wide adoption of data sanitization methods for protecting sensitive information, the exact impacts of redacting PII content from natural language data on the performance of language models has not been studied in-depth to the best of the authors' knowledge. Making inferences based on context is core to how language models function, and therefore stripping away contextual identifiers will likely reduce a model's ability to comprehend text, thereby leading to a decrease in performance. Additionally, replacing diverse PII with generic tokens, for instance replacing two different names with the same <NAME> token introduces ambiguity, making it harder for the model to differentiate between unique entities. An example of how redacting PII impacts a large language model's (LLM's) thinking process is depicted in Figure 1.

This work aims to perform an empirical analysis of the impact of data sanitization on the language understanding capabilities of both small and large language models. Our main contributions include:

- We perform a number of ablation experiments by fine-tuning small models like BART [22] and GPT-2 [30] across several benchmark NLP tasks to better understand the impact of PII redaction.

- We also conduct prompting experiments with popular large language models like Claude 3.5 Sonnet [1], Mistral 7B [17], and GPT-4o [27] to investigate the impact of PII redaction on some common generative artificial intelligence (GenAI) datasets. This analysis is first done using a full set of redacted entities, and subsequently on a limited set by not redacting task-critical entities.

- Finally, we investigate the correlation between task performance and number of named entities being redacted by using different sampling techniques, and suggest a strategy to best utilize an already redacted dataset without compromising on accuracy.

## 2 Related Work

### 2.1 Data Sanitization Tools

Data sanitization is generally achieved through replacing PII with non-sensitive tokens (e.g. <NAME> or <SSN>) prior to use. Santization techniques typically use sequence labeling approaches, such as named entity recognition (NER) algorithms, to identify potential PII entities, which are then replaced [9, 16]. One such open-source anonymization tool from Kleinberg and Mozes, called NETANOS (named entity-based text anonymization for open science) [19], uses the Stanford Named Entity Tagger [12] to identify PII in data and then replaces the entities with their types (e.g., a name is replaced with a [NAME] token).

A central criticism of data-sanitization-based data protection techniques is that they only redact PII. A newer data sanitization technique called Textwash [18] includes a tag that comprises a meta-category which encapsulates the potentially sensitive information (PSI) concept. Concretely, PSI refers to the full spectrum of textual information that could reveal an identity but cannot be attributed to a well-defined PII entity.

### 2.2 Training with Sanitized Data

Many related works have incorporated data sanitization into multi-step training pipelines. For example, Shi *et al.*[32] propose a two step fine-tuning method in which they train on redacted data in phase one, and again with the original unredacted data using a private training mechanism in phase two. Kong *et al.*[20] propose a systematic framework for redacting data from pre-trained generative models. They compare data redaction to data deletion and look at which data samples are 'hard' to redact. While there is abundant research on how to anonymize data [14, 28] and the associated privacy risks [25, 3, 29], there is limited research on the effects of PII redaction on model performance.

In the current era where large language models (LLMs) have made a pronounced impact across a number of fields of research, an analysis into the impact of data sanitization on their performance is of profound importance. Decisions about which LLMs to use for a task are often based on their performance on public benchmarks [6]. However, when these models are applied on domain-specific tasks with high amounts of sensitive entities redacted, their performances drastically drop at times. This paper aims to showcase this analysis across a number of popular generative artificial intelligence (GenAI) benchmarks.

## 3 Experimental Design

To thoroughly understand the impact of data sanitization on LM performance, we perform a number of experiments spanning small to large language models, and across a variety of NLP and GenAI datasets. To perform data sanitization we used the named entity recognition (NER) tool provided by the spaCy package [15]. The entities that we redact in this work are person names, locations, organization names, dates, times, sensitive encoded numbers such as credit card or social security numbers, email addresses, and intellectual property such as names of movies or various works of art.

### 3.1 Datasets

**Traditional NLP Datasets**: We begin our analysis on a set of traditional natural language processing tasks corresponding to the GLUE [35] and LexGLUE [5] benchmarks: semantic similarity with the Quora Question Pairs (QQP) dataset [8], textual entailment prediction on the Multi-Genre Natural Language Inference (MultiNLI) dataset [37], reading comprehension on the Winograd Schema Challenge dataset [21], multi-class classification on the LEDGAR dataset [34], and multi-label classification on the EURLEX dataset [5]. Additionally, we also consider tasks such as extractive question answering (Q&A) on the SQuADv2.0 dataset [31] and sentiment analysis on the IMDB movie review sentiment classification dataset [24]. These tasks are selected due to a substantial fraction of the total samples in each task containing PII entities. Specifically, all tasks contain at least 49% of samples with PII entities, and some tasks contain as much as 90% of samples with PII entities. Table 4 in Appendix 6.1 shows the full statistics on the composition of these datasets.

**GenAI Datasets**: We include the following recently released datasets that are used to benchmark modern large language model (LLM) performances in our analysis: Discrete reasoning over content of paragraphs on DROP [11] dataset, linguistically diverse grade school math word problems on GSM8K [7], and a set of tasks from Big-Bench-Hard (BBH) benchmark [33]. For BBH, we excluded some tasks which did not have any PII entity, or contained non-english words. Details of the inclusion/ exclusion criteria for BBH tasks is provided in Appendix 6.1.2. We also retained the SQuADv2.0 and IMDB datasets from our earlier study here, as they are still popular in the LLM community.

For both sets of datasets, we redacted the entities mentioned in Section 3 using the spaCy tool. Since the SQuADv2.0 dataset requires computing the span of the answer, in other words the indices at which the answer begins and ends, we modified the indices in the train and dev sets based on the length of the redaction token to ensure that the answer spans remain consistent.

## 3.2 Language Models

To separately study the impact of data sanitization across fine-tuning and prompting tasks, we performed our analysis using a host of small and large language models.

**Fine-Tuning with small language models**: We consider models with <5B parameters in this category. For this study, we investigate the effects of data sanitization across BART [22] and GPT-2 [30] models. We selected these models to cover both encoder and decoder blocks (BART) and decoder-only blocks (GPT-2). We have fine tuned these models on the NLP datasets discussed in Section 3.1, and evaluated the performance on several different train/test dataset pairings as mentioned in 4.1.

**Prompting large language models**: For large language models (LLMs) with >5B parameters, we have used chain-of-thought (CoT) prompting with few-shot examples [36] to study the effects of data sanitization on the following models: Anthropic's Claude 3.5 Sonnet [1], Mistral AI's Mistral 7B [17], and OpenAI's GPT 4o [27]. It should be noted that for both the original and redacted evaluation sets, the few-shot examples used here were from the original unredacted train set, under the hypothesis that prior to evaluation in many real-world applications on sanitized, we may not have access to the test data. The results for these experiments are found in Section 4.2.

# 4   Results

In this section, we discuss the results of our various experiments. We start with the analysis on fine-tuning smaller language models in Section 4.1, followed by discussion on prompting large language models in Section 4.2.

## 4.1   Fine-Tuning Results

We present the evaluation results for transformer models trained with both sanitized and unsanitized data for the benchmark NLP tasks discussed in Section 3.1. Within each of these tasks, we calculated performance metrics for the following train/test dataset pairings[1]:

- None/None: The performance of the model trained on unredacted data and evaluated on unredacted data. This is our baseline result where no PII redaction is applied.

- Redact/Redact: The performance of the model trained on redacted data and evaluated on redacted data. This is the most conservative redaction policy, where all PII is redacted for both training and inference. Such a redaction policy may be required when inference is performed in batch.

- None/Redact: The performance of the model trained on unredacted data and evaluated on redacted data. This may be applicable if third party models are applied to redacted data.

In cases where the $test$ set label is not provided for independent evaluation, we tested on the labeled $dev$ set.

**Model and Task Comparison**: The performance comparison of redaction on BART and GPT-2 models across NLP tasks is provided in Table 1. For all datasets, we observe that the performance impact between None/None and Redact/Redact is less as compared to None/None and None/Redact.

---

[1]We omitted the Redact/None pair from our analysis as we do not foresee any real-world use-case for it.

Table 1: Performance results on NLP datasets. For each dataset, the model performances are shown for different combinations of original and redacted versions across training and validation splits. All numbers are in %.

| Datasets | BART | | | GPT-2 | | |
|---|---|---|---|---|---|---|
| | None/None | Redact/Redact | None/Redact | None/None | Redact/Redact | None/Redact |
| *Low impact (<10%)* | | | | | | |
| IMDB (Acc) | 93.7 | 93.7 | 93.6 | 93.1 | 93.2 | 92.7 |
| LexGLUE: LEDGAR (F1) | 87.0 | 87.0 | 85.7 | 87.5 | 87.0 | 85.8 |
| LexGLUE: EURLEX (F1) | 66.3 | 66.3 | 65.2 | 64.1 | 62.1 | 60.1 |
| GLUE: MNLI (m) (Acc) | 85.9 | 83.7 | 81.7 | 81.8 | 81.5 | 77.8 |
| GLUE: MNLI (mm) (Acc) | 86.1 | 84.3 | 82.5 | 82.5 | 82.0 | 79.1 |
| GLUE: WNLI (Acc) | 47.9 | 47.9 | 47.9 | 47.0 | 47.0 | 47.9 |
| GLUE: QQP (Acc) | 90.4 | 88.5 | 83.8 | 89.0 | 86.9 | 82.9 |
| *Moderate impact (10-25%)* | | | | | | |
| SQuADv2.0 (F1) | 74.9 | 57.7 | 60.2 | 55.8 | 48.7 | 48.7 |

This is likely due to the misalignment between the training and the test sets when redaction is introduced to the model. Another observation is that the results suggest only minimal degradation in model performance when training on redacted data, with model performance decreasing less than 2.2% on the average. The models are robust to these tasks and in some cases (for instance in the case of IMDB and LexGLUE) no impact is observed because for these tasks the language models needs the entire context to make a prediction rather than relying on specific PII entities.

An exception to the above mentioned observation is seen for GLUE: QQP and SQuADv2.0 datasets. For QQP, even though it is classified as low impact the performance difference between None/None and None/Redact is high when compared to other tasks (7% vs 2% on average). A possible reason for this is that the redaction of PII entities may cause a language model to pay more attention on surrounding context rather than query-critical PII entities, potentially leading to inaccurate semantic similarity scores. Additionally, SQuADv2.0 is an extractive question answering task that suffers the largest degradation in model performance when trained on redacted data. We hypothesize that this is an expected result for models trained on Q&A tasks, which are likely more reliant on leveraging contextual PII entities and references to identify answers than models trained for other tasks, such as sentiment analysis or entailment.

## 4.2 Prompting generative model Results

As many of the popular large language models are not open-source, we do not have the flexibility to fine-tune them on a redacted set. In lieu of that, we discuss the impact of PII redaction on performance of large language models that are *prompted* to solve a task. This is similar to the None/Redact setup that we used for the smaller models in Section 4.1. Table 2 provides the scores for Claude 3.5 Sonnet, Mistral 7B, and GPT 4o on the GenAI datasets mentioned in Section 3.1. For each dataset, we report the performance on the original (None) and redacted (Redact) sets, along with the relative impact.

### 4.2.1 General Observations

**Low vs Medium vs High impact datasets**: We observe that the impact of redaction on the different tasks range from 0.3% to 95% for Claude, -2.7% to 67.3% for Mistral and -6.5% to 100% for GPT. Based on these results, we have classified the datasets as low impact if the impact on performance was < 10%, medium impact if the impact on performance was between 10 and 25% and high impact for those datasets where the impact was greater than 25%. If a dataset such as BBH: formal fallacies has diverse impact across the three models, then we based our classification on the majority vote. One observation is that the severity of the impact is not only dependant on the number of records redacted in a dataset but also on the type of the task and how important the entities are to reason about a given task e.g. BBH: causal judgement had 91% of the dataset redacted but is still a low impact dataset as the correct answer in that task is dependant on the entire context provided to the LLM rather than the

Table 2: Performance results on GenAI datasets. For each dataset, the model performances are shown on the original and redacted versions, along with their relative impact. All numbers are in %.

| Datasets | Claude 3.5 Sonnet | | | Mistral 7B | | | GPT 4o | | |
|---|---|---|---|---|---|---|---|---|---|
| | None | Redact | Impact | None | Redact | Impact | None | Redact | Impact |
| *Low impact (<10%)* | | | | | | | | | |
| IMDB | 95.8 | 95.5 | 0.3 | 86.5 | 86.6 | -0.1 | 93.9 | 93.1 | 0.9 |
| BBH: Hyperbaton | 99.6 | 99.2 | 0.4 | 49.6 | 49.2 | 0.8 | 100 | 98.8 | 1.2 |
| BBH: Disambiguation QA | 75.5 | 74.0 | 2.0 | 58.8 | 60.4 | -2.7 | 80.0 | 85.2 | -6.5 |
| BBH: Snarks | 90.4 | 88.1 | 2.5 | 52.2 | 48 | 8.04 | 89.8 | 87 | 3.2 |
| BBH: Ruin Names | 90.4 | 84.3 | 6.7 | 34 | 32.8 | 3.52 | 90.8 | 85.6 | 5.7 |
| BBH: Logical Deduction (#3) | 98.7 | 91.2 | 7.6 | 46.4 | 41.2 | 11.2 | 100 | 90.0 | 9.6 |
| BBH: Causal Judgement | 69.0 | 63.0 | 8.7 | 42.8 | 42.2 | 1.35 | 67.0 | 65.0 | 2.7 |
| BBH: Formal Fallacies | 88.0 | 75.0 | 14.8 | 60 | 57.2 | 4.7 | 78.0 | 74.0 | 5.6 |
| *Moderate impact (10-25%)* | | | | | | | | | |
| SQuADv2.0 | 65.8 | 57.8 | 12.2 | 46.1 | 30.5 | 33.8 | 68.3 | 51.4 | 24.7 |
| BBH: Logical Deduction (#5) | 93.6 | 82.7 | 11.6 | 24.4 | 26.0 | -6.5 | 91.6 | 80 | 12.6 |
| BBH: Logical Deduction (#7) | 83.5 | 64.7 | 22.6 | 22.8 | 18.4 | 19.3 | 79.6 | 66.8 | 16.1 |
| *High impact (>25%)* | | | | | | | | | |
| DROP | 92.1 | 54.2 | 41.1 | 46.1 | 25.9 | 43.8 | 91.6 | 49.3 | 46.2 |
| GSM8K | 96.9 | 44.6 | 54.0 | 45.3 | 19.0 | 58.1 | 57.6 | 25.5 | 55.7 |
| BBH: Date Understanding | 92.8 | 40.6 | 56.3 | 40.4 | 13.2 | 67.3 | 90.8 | 19.6 | 78.4 |
| BBH: Penguins in a Table | 99.3 | 30.8 | 69 | 43.8 | 29.4 | 32.9 | 99.0 | 47.0 | 48.2 |
| BBH: Tracking Shuffled Objects (#5) | 100.0 | 8.8 | 91.2 | 27.6 | 18 | 34.8 | 98.4 | 4 | 95.9 |
| BBH: Tracking Shuffled Objects (#7) | 100.0 | 4.8 | 95.2 | 22.8 | 12.8 | 43.9 | 99.6 | 0 | 100 |

entity value. This also explains why the BBH tasks are split across low, medium and high impact datasets.

**Oddities in Mistral's performance on Redacted Datasets**: A consistent observation from Table 2 is that the Claude 3.5 and GPT 4o models tend to perform similarly in terms of redaction impact, while Mistral demonstrates some uncommon properties. This happens because Mistral has a tendency to hallucinates and assign placeholder values for redacted entities, and then reason about them incorrectly to arrive at the correct answer. An example of this is depicted in Figure 2. The extreme manifestation of this type of hallucination is seen for IMDB, BBH: logical deduction (#5), and BBH: disambiguation qa datasets, where the performance on the redaction set is even better than the original unredacted set.

### 4.2.2 Weaker Redaction for High Impact Datasets

In the previous section we have discussed the redaction impact on language models by covering the full set of entities redacted as mentioned in Section 3. This leads to the low, moderate, and high impact clusters of datasets. Upon diving deep into the high impact datasets, we observed that some of the entities redacted are too harsh, and removing them often makes the dataset impossible to solve. An example of this is redacting the <DATE> entity in BBH: date understanding task,

Table 3: Performance results for limited redaction across tasks using Claude 3.5 Sonnet

| Datasets | Redaction Amount | | | Limited Redaction | Redacted |
|---|---|---|---|---|---|
| | None | Full | Limited | Impact | PII Entities |
| DROP | 92.1 | 54.2 | 79.3 | 13.8 | NAME, LOC, ORG |
| GSM8K | 96.9 | 44.6 | 90.1 | 7.0 | NAME, LOC, ORG |
| BBH: Date Understanding | 92.8 | 40.6 | 86.3 | 7.0 | NAME |
| BBH: Penguins in a Table | 99.3 | 30.8 | 82.9 | 16.5 | NAME, ORG |
| BBH: Tracking Shuffled Objects (#5) | 100.0 | 8.8 | 95 | 5 | LOC, ORG, DATE |
| BBH: Tracking Shuffled Objects (#7) | 100.0 | 4.8 | 93 | 7 | LOC, ORG, DATE |

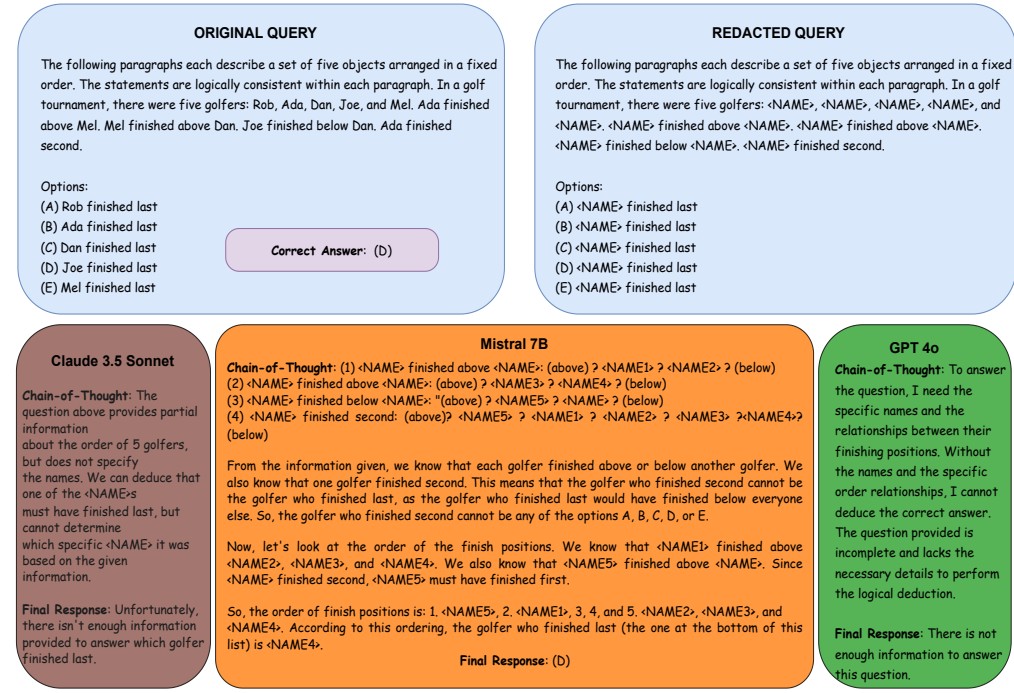

Figure 2: Mistral's hallucination in the context of entity redaction. As shown in the figure, Claude and GPT models correctly highlight the lack of information present in the query due to redaction, and proceed to not provide any answer. In contrast, Mistral assigns sequential values to various <NAME> tags, and reasons about them to arrive to the correct answer. This explains the trend of Mistral's performance getting less impacted by redaction as compared to the other models.

which results in a huge drop in performance as expected. To investigate whether excluding some task-critical entities from the redaction set can restore the full accuracy of these impacted tasks, we run another experiment with weaker redaction. Our results are presented in Table 3, where for each dataset, we show the limited set of PII entities that we redacted. We run these experiments only for the Claude model since that was the best performing among all the three models we tested. With weaker redaction, we observe that nearly all of the high impact tasks can now be re-classified as low impact as per our thresholds. The exceptions are DROP and BBH: penguins in a table, which are still moderate impact. We hypothesize that for these two datasets, there is at least one non-critical PII entity (<QUANTITY> for the penguins task and <NAME> and <ORG> for DROP) which is dominant, causing its performance to still not be comparable to the unredacted dataset's performance.

### 4.2.3 Correlation between number of Redacted Entities and Performance

Given that it is extremely difficult to guarantee that a provided test dataset has been completely sanitized, an evident detail to investigate is the correlation between task performance and number of named entities (NE) being redacted. For this, we performed a systematic experiment where we progressively redacted parts of the dataset, and evaluated the models' performances. Naturally, there are two ways to choose the number of samples to redact, (i) random and (ii) based on named entity (NE) content. The plot in Figure 3 shows the relative performance between these sampling options. It is interesting to see that if we randomly redact samples, there is a linear decline as expected. However, with content sampling, we can see better performance on the dataset when redacting low content entities (see Figure 3). Moreover, for DROP dataset, the correlation between performance drop and number of redacted entities is not that strong (the lines are collinear with random dropping). We hypothesize this to be related to the presence of a diverse set of PII entities present in this dataset. Please refer to Appendix 6.1.1 for more details about this.

### 4.2.4 Strategy to repair a given Redacted Dataset

In many real-world applications involving GenAI algorithms, developers often do not have control over the degree of redaction within the dataset, and have to make the best possible use of it in its

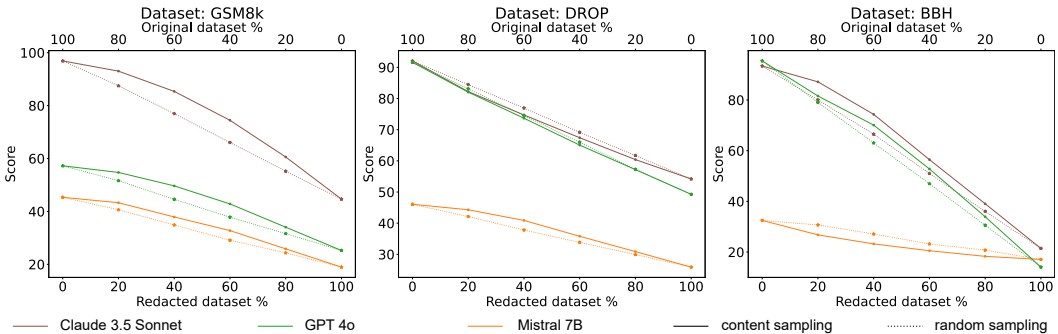

Figure 3: Performance of random vs content sampling with replacement for all high-impact datasets. The trend shows that randomly redacting a portion of the dataset leads to a linear drop in performance, whereas by redacting samples based on the PII content leads to a non-linear drop. This non-linearity trend is more prominent for GSM8k and BBH datasets, while less for DROP dataset. We hypothesize the reason to be related to a more uniform distribution of PII content in DROP dataset, thereby making the sampling methods equivalent.

redacted state. A prime example of this is in the domain of handling customer-centric data. For these use-cases, there is a strong need of developing a strategy to filter out portions of the data which can impact performance. One such strategy involves subsampling a given redacted dataset by removing high PII-content records, and using the remaining ones. This is shown in Figure 4. As seen for the SQuADv2.0 and GSM8k datasets, there is a wide difference in performance as we filter out records based on content as opposed to randomly. An interesting exception to this is DROP dataset, which does not follow the trend. This is possibly related to the fact that as that dataset contains a large amount of text, greedily filtering out high content records might not always be detrimental to the main task. An illustration of this is shown in the Appendix 6.1.1.

## 5   Conclusions & Future Work

In this paper we have demonstrated that on smaller language models, training data sanitization has minimal impact on model performance across most benchmark NLP tasks, with extractive question answering being one notable exception. For all other tasks studied, we observe less than a 2.5% drop in performance, which may be tolerable in production settings to meet data sanitization requirements. When prompting LLMs, we find that GPT 4o and Claude 3.5 have similar impact with respect to redaction, while Mistral 7B demonstrates some interesting exceptions. Furthermore, this impact varies by task, and for high impact tasks, care is necessary in selecting the entities to redact. Some possible directions of future research include investigating further into specific domains of datasets such as medical or legal where PII presence is strong, experimenting with diverse redaction techniques such as pseudo-anonymization, and development of better PII redaction tools.

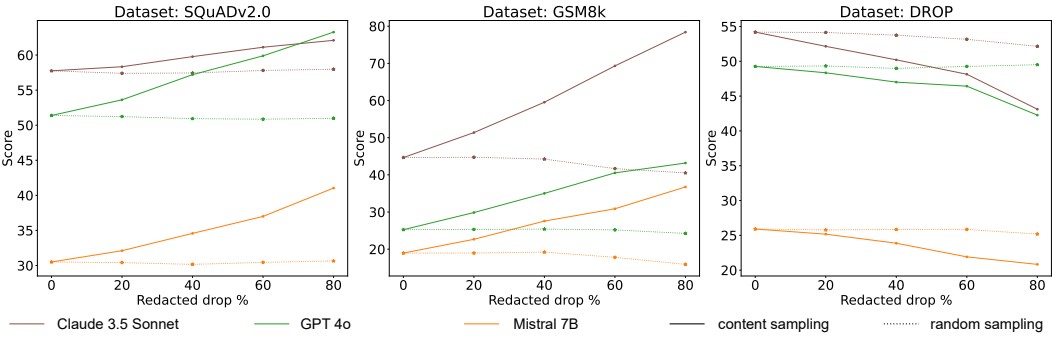

Figure 4: Performance of LLMs on a fraction of the dataset obtained by random vs content sampling. The trend shows that for SQuADv2.0 and GSM8k datasets, it is possible to *repair* these datasets by removing samples that are heavily redacted. Interestingly, DROP does not follow this trend. We hypothesize this to be due to the *uniformly diverse* PII present content there, ensuring that simply by removing samples based on the count does not ensure performance improvement.

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

Table 4: Dataset redaction statistics

| Datasets | Percentage of Redacted Dataset |
|---|---|
| DROP | 100.00 |
| GSM8K | 89.01 |
| GLUE - QQP (train/dev) | 51.73/52.15 |
| GLUE - MNLI (train/dev m/mm) | 1.51/52.47/49.39 |
| GLUE - WNLI (train/dev) | 68.19/52.11 |
| SQuADv2.0 (train/dev) | 94.46/91.17 |
| IMDB (train/test) | 85.55/81.60 |
| LexGLUE LEDGAR (train/test) | 85.55/81.60 |
| LexGLUE EUR-LEX (train/test) | 100.00/100.00 |

[36] Jason Wei, Xuezhi Wang, Dale Schuurmans, Maarten Bosma, Fei Xia, Ed Chi, Quoc V Le, Denny Zhou, et al. Chain-of-thought prompting elicits reasoning in large language models. *Advances in neural information processing systems*, 35:24824–24837, 2022.

[37] Adina Williams, Nikita Nangia, and Samuel Bowman. A broad-coverage challenge corpus for sentence understanding through inference. In *Proceedings of the 2018 Conference of the North American Chapter of the Association for Computational Linguistics: Human Language Technologies, Volume 1 (Long Papers)*, pages 1112–1122. Association for Computational Linguistics, 2018.

[38] David Williams. Systems and methods for automatically scrubbing sensitive data, May 9 2023. US Patent 11,645,458.

# 6 Appendix

## 6.1 Dataset statistics

In this section we provide the percentage of samples in which a given entity type was redacted for the datasets mentioned in Section 3.2. The statistics for all datasets is provided in Table 4 while the BBH dataset statistics are provided in Table 5.

### 6.1.1 Diverse PII content in DROP dataset

As mentioned in Sections 4.2.2, 4.2.3 and 4.2.4, the DROP dataset demonstrates some unique characteristics in the form of larger impact despite limited redaction, lesser correlation between number of redacted entities and impact, and also difficulty in restoring performance despite content-based sample dropping. This lead us to inspect deep into this dataset, and we observed that as there are longer passages here, the chances of PII content is also high. However, not all of the entities are task-critical, and hence simply not redacting few entities, or various content-based sampling strategies might not help boost the performance. An illustration of this is shown in Figure 5.

### 6.1.2 Inclusion / Exclusion criteria for BBH tasks

For the BBH dataset, we only chose a small subset of tasks based on the following exclusion criteria:

- Any dataset in which the task is to reason about a single dominant PII entity such as name or date whose exclusion will render the dataset impossible to reason about even by a human being. For these datasets, we hypothesised that not redacting that single dominant entity will have results similar to no redaction results. In Table 6 we demonstrate this by showing that the limited redaction performance is similar to no redaction. This excludes the following datasets: temporal sequences, tracking shuffled objects (#3), movie recommendations, sports understanding, web of lies. For all these datasets the single dominant PII entity was either name or date. While this criteria excludes tracking shuffled objects (#3), it does not excludes tracking shuffled objects (#5) and tracking shuffled objects (#7) as both those dataset have diverse PII entity rather than one dominant PII

Table 5: BBH Dataset redaction statistics

| BBH Datasets | Percentage of Redacted Dataset |
|---|---|
| Hyperbaton | 4 |
| Disambiguation QA | 14.4 |
| Snarks | 24.7 |
| Ruin Names | 20 |
| Causal Judgement | 91.4 |
| Logical Deduction (#3) | 25.2 |
| Logical Deduction (#5) | 26 |
| Logical Deduction (#7) | 50.8 |
| Date Understanding | 100 |
| Penguins in a Table | 100 |
| Tracking Shuffled Objects (#3) | 100 |
| Tracking Shuffled Objects (#5) | 100 |
| Tracking Shuffled Objects (#7) | 100 |
| Temporal Sequences | 100 |
| Formal Fallacies | 100 |
| Web Of Lies | 100 |
| Multistep Arithmetic Two | 0 |
| Sports Understanding | 96.4 |
| Word Sorting | 0 |
| Movie Recommendation | 100 |
| Salient Translation Error Detection | 100 |
| Geometric Shapes | 0 |
| Reasoning About Colored Objects | 90.8 |
| Boolean Expressions | 0 |
| Dyck Languages | 0 |
| Navigate | 0 |
| Object Counting | 0 |

entity. An example is demonstrated in Figure 6 which shows that there are two type of PII entities which are required to reason about the question rather than one dominant PII entity.

- Any dataset that had 0 PII entities was excluded from our experiments. These included dyck, boolean expressions, navigate, object counting, multistep arithmetic two, word sorting and geometric shapes dataset.

- Any dataset that had non-english words were excluded as the PII redaction tool (spaCy) is for English Language. This excluded the salient translation error detection dataset.

Table 6: Performance results for limited redaction across tasks using Claude 3.5 Sonnet

| Datasets | Redaction Amount | | | Redacted PII Entities |
|---|---|---|---|---|
| | None | Full | Limited | |
| BBH: Temporal Sequences | 99.6 | 2.4 | 97.6 | NAME, ORG |
| BBH: Tracking Shuffled Objects (#3) | 100.0 | 25.4 | 98 | LOC, ORG, DATE |
| BBH: Web Of Lies (#3) | 100.0 | 44 | 100 | None (Name was the only entity in web of lies) |

**REDACTED QUERY**

**Passage**: For <DATE>, the <NAME> traveled to <LOC> to take on the <LOC> <ORG>. It also marked <DATE> that the <NAME> began <DATE> on the road. On <DATE> before the first game of <DATE>, the <ORG> placed <NAME> on injured reserve due to an injured back. They signed <NAME> off the practice squad to do the kicking. The <ORG> started well as the <ORG> tried to surprise the <ORG> with an onside kick on the opening kickoff which the <ORG> recovered. Four plays later, <NAME> hit <NAME> who bowled into the endzone for an eight-yard touchdown pass to give the <ORG> the lead 7-0. The <ORG> answered on their ensuing drive, going <QUANTITY> on 12 plays with <ORG> rushing for a 10-yard touchdown to tie the game. The <ORG> came back as <LOC> hit his first career field goal, a 20-yarder, to give the <ORG> a 10-7 lead. After exchanging punts in the early part of <DATE>, the <ORG> tied the game with <TIME> remaining in the half on a <NAME> 23-yard field goal. The <ORG> offense moved <QUANTITY> with time winding down to set up <LOC>'s second field goal, from <QUANTITY> out, to give the <ORG> the halftime lead 13-10. In <DATE>, the <ORG> defense held the <ORG> to punts on consecutive drives as the <ORG> notched another 52-yard field goal to extend the lead to 16-10 with a little over four minutes left in <DATE>. The <ORG> defense prevented a <ORG> touchdown on the ensuing drive, forcing the <ORG> to settle for a 26-yard field goal which narrowed the gap to 16-13 as the period expired. In <DATE>, <NAME> led the <ORG> on a seven play, 70-yard drive capped off by a <NAME> 19-yard touchdown reception. The <NAME> defense, which had struggled immensely <DATE>, forced the <ORG> to punt and then held them to a field goal with <TIME> remaining. Another <LOC> field goal, this time from <QUANTITY> out, extended the lead to 26-16 with 1:14 left in the game. The <ORG> could muster no more, missing a 52-yard field goal with <TIME> remaining as the <ORG> pulled out the win 26-16. In the win, <LOC> became the first kicker in <ORG> history to make two 50-yard field goals in an <ORG> debut. Following his performance, <LOC> was named <ORG> of the Week. <LOC> ran for <QUANTITY> on 18 carries in his <ORG> debut. <NAME> threw for <QUANTITY> with two touchdown passes to lead the <ORG> to a 1-0 record to start <DATE>.

**Question**: How many more yards was the longest field goal over the shortest one?

**Chain-of-Thought**

The question is asking about the difference in yards between the longest and shortest field goals mentioned in the passage. I need to find all the field goal distances mentioned and compare them. Looking through the passage, I find the following field goal distances:

1. 20-yard field goal
2. 23-yard field goal (by <NAME>, not the <ORG>'s kicker)
3. 52-yard field goal
4. 26-yard field goal (by the <ORG>, not the <ORG>' kicker)
5. 43-yard field goal

Considering only the field goals by the <ORG>' kicker, the longest is 52 yards and the shortest is 20 yards.

To calculate the difference: 52 - 20 = 32 yards. Therefore, the longest field goal was 32 yards longer than the shortest one.

**Final Response**: 32 ✅

**Observation**: In the provided passage, there are a diverse number PII entities present => {'DATE': 11, 'NAME': 11, 'LOC': 8, 'ORG': 29, 'QUANTITY': 6, 'TIME': 3}. Despite this, none of the entities are critical to the question asked, and therefore the language model is able to correctly reason about the given problem and arrive at the right answer. This is an illustration of the fact that for DROP dataset, there is not a strong correlation between large number of entities, and lower model performance.

Figure 5: [Best viewed in color] The figure illustrates that DROP dataset has a diverse number of PII entities present, but that does not necessarily impact performance when the question asks about a specific unredacted portion of a long redacted passage.

**ORIGINAL QUERY**

Alice, Bob, Claire, Dave, Eve, Fred, and Gertrude are friends and avid readers who occasionally trade books. At the start of the semester, they each buy one new book: Alice gets Lolita, Bob gets Frankenstein, Claire gets Moby Dick, Dave gets The Pearl, Eve gets Hound of the Baskervilles, Fred gets The Fellowship of the Ring, and Gertrude gets The Odyssey. As the semester proceeds, they start trading around the new books. First, Dave and Gertrude swap books. Then, Claire and Bob swap books. Then, Alice and Dave swap books. Then, Fred and Claire swap books. Then, Dave and Fred swap books. Then, Bob and Claire swap books. Finally, Eve and Fred swap books. At the end of the semester, Eve has

Options:
(A) Lolita
(B) Frankenstein
(C) Moby Dick
(D) The Pearl
(E) Hound of the Baskervilles
(F) The Fellowship of the Ring
(G) The Odyssey

**REDACTED QUERY**

<NAME>, <NAME>, <NAME>, <NAME>, <NAME>, <NAME>, and <NAME> are friends and avid readers who occasionally trade books. At the <DATE>, they each buy one new book: <NAME> gets <NAME>, <NAME> gets <NAME>, <NAME> gets <WORK_OF_ART>, <NAME> gets <WORK_OF_ART>, <NAME> gets <WORK_OF_ART>, <NAME> gets <WORK_OF_ART>, and <NAME> gets <WORK_OF_ART>. As the semester proceeds, they start trading around the new books. First, <NAME> and <NAME> swap books. Then, <NAME> and <NAME> swap books. Then, <NAME> and <NAME> swap books. Then, <NAME> and <NAME> swap books. Then, <NAME> and <NAME> swap books. Then, <NAME> and <NAME> swap books. Finally, <NAME> and <NAME> swap books. At the <DATE>, <NAME> has

Options:
(A) <NAME>
(B) <NAME>
(C) <WORK_OF_ART>
(D) <WORK_OF_ART>
(E) <WORK_OF_ART>
(F) <WORK_OF_ART>
(G) <WORK_OF_ART>

Figure 6: [Best viewed in color] Diverse PII entities present in the BBH: Tracking Shuffled Objects (#7) shows that simply not redacting the dominant entity may not recover the performance fully.

