# OpenReview forum: "The Empirical Impact of Data Sanitization on Language Models"
_NeurIPS.cc/2024/Workshop/SafeGenAi — SafeGenAi Poster_

### Official Review · Reviewer_RHvc · 2024-10-08
**Review Comments of the Empirical Impact of Data Sanitization on Language Models**

**Rating:** 7
**Confidence:** 2

**Review:**

This paper empirically analyzes the effects of data sanitization on language models (LLMs) across various benchmark tasks, including comprehension question answering, entailment, sentiment analysis, and text classification. The authors investigate the performance impact of redacting personally identifiable information (PII) and other sensitive entities from the training datasets. They also explore the correlation between task performance and the number of redacted named entities, and propose a strategy to repair already redacted datasets using content-based subsampling.
Strengths:
The paper presents a thorough analysis of the impact of data sanitization on LLMs across a wide range of NLP and GenAI benchmark tasks.
The authors conduct a series of experiments using different language models and datasets, providing a robust evaluation of the effects of data sanitization.
The paper reveals that data sanitization has minimal impact on model performance for most tasks, but significantly affects extractive question answering. Additionally, the severity of the impact depends on both the number and type of redacted entities.
Weaknesses:
The findings of this paper may not generalize to all types of LLMs and tasks, as the experiments were conducted on a specific set of models and datasets.
In conclusion, this paper makes a significant contribution to the understanding of the impact of data sanitization on LLMs by presenting a comprehensive empirical analysis across various benchmark tasks. While the findings are insightful, the authors should consider expanding the experimental scope, simplifying the repair strategy, and providing a theoretical analysis to further enhance the paper's quality and impact.





In conclusion, the paper presents an interesting study on the empirical impact of data sanitization on language models' language understanding capabilities. However, it could benefit from more detailed descriptions of the methodology, datasets, and results analysis. Additionally, expanding the discussion section to include implications, limitations, and future research directions would enhance the paper's contribution to the field.

---

### Official Review · Reviewer_v6p2 · 2024-10-08
**Review for The Empirical Impact of Data Sanitization on Language Models**

**Rating:** 6
**Confidence:** 5

**Review:**

This paper investigates the effects of data sanitization on language model performance across various NLP tasks, revealing that the impact of redacting personally identifiable information (PII) varies by task, with some tasks like comprehension question answering experiencing significant performance drops, while others are minimally affected. While the motivation of paper is clear, its novelty may be lacking. A few comments below:
1. There are quite a few redaction techniques including pseudo-anonymization and entity masking. Will the authors consider comparing performance with other techniques too?
2.  The large performance drop (>25%) in comprehension question answering (Q&A) can be concerning.
3. There is emphasis on smaller models, will the authors find relevance in large models too?

---

### Official Review · Reviewer_am4R · 2024-10-09

**Rating:** 6
**Confidence:** 4

**Review:**

**Summary**

This paper addresses the underexplored issue of how data sanitization, specifically redacting personally identifiable information (PII), affects the performance of language models (LMs) across various natural language processing (NLP) tasks. The authors conducted a series of experiments to examine the performance degradation that results from redacting different types of entities across both small and large language models. They provide valuable empirical results on tasks such as sentiment analysis, question answering, and entailment prediction, among others. The paper makes an important contribution to understanding the trade-offs between privacy protection and model performance.

**Strength**

- This work addresses a novel and relevant problem: the empirical impact of data sanitization on language models, particularly in the context of maintaining data privacy. While data sanitization is a widely recognized method for privacy, its performance implications have been underexplored, making this paper a valuable contribution to the field.
- The paper follows a structured experimental methodology, clearly defined in the introduction and design sections. It rigorously tests both small and large language models on a variety of tasks, including traditional NLP and Generative AI (GenAI) benchmarks. The empirical findings, particularly on the differences in performance degradation across tasks (e.g., question answering vs. sentiment analysis), are insightful and well-supported by the data.
- The analysis provides meaningful takeaways for real-world applications where privacy is critical. For example, understanding that sentiment analysis is more resilient to PII redaction compared to question answering is useful for designing privacy-aware systems.
- The discussion around "weaker redaction" strategies (i.e., selectively redacting entities based on task importance) is especially practical, adding depth to the proposed solutions for mitigating performance loss.

**Weakness**

- The paper could benefit from a more detailed explanation of the redaction techniques and the differences between the redaction approaches used across datasets (e.g., how varying types of PII affect different tasks).
- The paper’s abstract could do a better job of highlighting the main findings. Specifically, summarizing the magnitude of performance degradation for different tasks and models would make the abstract more informative.
- The originality of the proposed redaction repair strategy could be further developed. While the idea of weaker redaction is useful, it is largely an empirical observation.
- The paper would be more impactful if it included a more direct discussion of how the findings could be applied to privacy regulation compliance, or if it provided recommendations for privacy-preserving language model development.